# Exogenous Integrin αIIbβ3 Inhibitors Revisited: Past, Present and Future Applications

**DOI:** 10.3390/ijms22073366

**Published:** 2021-03-25

**Authors:** Danique L. van den Kerkhof, Paola E.J. van der Meijden, Tilman M. Hackeng, Ingrid Dijkgraaf

**Affiliations:** Department of Biochemistry, Cardiovascular Research Institute Maastricht (CARIM), Maastricht University, 6229 ER Maastricht, The Netherlands; d.vandenkerkhof@maastrichtuniversity.nl (D.L.v.d.K.); p.vandermeijden@maastrichtuniversity.nl (P.E.J.v.d.M.); t.hackeng@maastrichtuniversity.nl (T.M.H.)

**Keywords:** platelets, integrin αIIbβ3, antagonists, hematophagous parasites

## Abstract

The integrin αIIbβ3 is the most abundant integrin on platelets. Upon platelet activation, the integrin changes its conformation (inside-out signalling) and outside-in signalling takes place leading to platelet spreading, platelet aggregation and thrombus formation. Bloodsucking parasites such as mosquitoes, leeches and ticks express anticoagulant and antiplatelet proteins, which represent major sources of lead compounds for the development of useful therapeutic agents for the treatment of haemostatic disorders or cardiovascular diseases. In addition to hematophagous parasites, snakes also possess anticoagulant and antiplatelet proteins in their salivary glands. Two snake venom proteins have been developed into two antiplatelet drugs that are currently used in the clinic. The group of proteins discussed in this review are disintegrins, low molecular weight integrin-binding cysteine-rich proteins, found in snakes, ticks, leeches, worms and horseflies. Finally, we highlight various oral antagonists, which have been tested in clinical trials but were discontinued due to an increase in mortality. No new αIIbβ3 inhibitors are developed since the approval of current platelet antagonists, and structure-function analysis of exogenous disintegrins could help find platelet antagonists with fewer adverse side effects.

## 1. Introduction

Integrins are an important class of transmembrane glycoprotein (GP) receptors consisting of an alpha- and a beta subunit, both of which have an extracellular domain, a transmembrane domain and a cytoplasmic domain. In addition to mediating communication between cells and the extracellular matrix (ECM), integrins enable cell-cell interactions. Integrins, therefore, play a major role in cell adhesion, survival and differentiation [1,2].

The integrin family members can be assembled from 18 α- and 8 β-subunits, which together can form 24 integrins with different effects on the function of cells. The α-subunits can be divided into two groups: subunits with an inserted domain (αI) and subunits without an inserted domain. This domain has a metal ion-dependent adhesion site (MIDAS), which allows ligands to bind the receptor [3]. Integrins are often expressed in the inactive state, although the binding affinity for ligands can be increased by the addition of Mn^2+^. The presence of Ca^2+^ in the blood has an essential role in keeping integrins in their inactive state [4].

The 9 α-subunits that contain the inserted domain are part of the collagen receptors (α1, α2, α10, α11) and leukocyte receptors (αE, αL, αM, αD, αX) [5,6]. The 9 subunits without the inserted domain can be classified into four categories based on evolution and specificity: (A) Arg-Gly-Asp (RGD) sequence receptors (αIIb, αv, α5, α8), (B) laminin receptors (α3, α6, α7), (C) PS3 (only found in insects), and (D) α4, α9 subunits [7].

Of the 8 different β-subunits that are found in integrins (β1-β8), two, β1 and β3, are found on platelets. The β1 integrins (α2β1, α5β1, and α6β1) support platelet adhesion to collagen, fibronectin and laminin (1000 copies per platelet). The αvβ3 and αIIbβ3 are β3 integrins, where αIIbβ3 (GPIIb/IIIa, CD41/CD61) is the most abundant integrin on platelets [1,8,9]. αvβ3 is found on smooth muscle cells and endothelial cells but is less abundant on platelets (50–100 copies per platelet) [10]. 

In this review, we will focus on integrin αIIbβ3, integrin αIIbβ3 inhibitors found in hematophagous parasites and snakes, the current inhibitors used in the clinic and those failed in clinical trials.

## 2. Integrin αIIbβ3GPIIb/IIIa

Integrin αIIbβ3–GPIIb/IIIa has approximately 80,000 copies per platelet and is crucial for haemostasis. Patients with Glanzmann thrombasthenia, a rare autosomal recessive disorder caused by mutations in genes encoding GPIIb and IIIa, suffer from bleeding episodes [11,12]. On the other hand, malfunction of integrin αIIbβ3 may contribute to thrombotic complications, such as ischemic stroke and myocardial infarction [13]. A partial deletion of the β3 in a megakaryocyte will lead to abnormally large platelets, (macro)thrombocytopenia (platelets < 150 · 10^9^/L blood) and bleeding tendencies [14].

The α-subunit of αIIbβ3 consists of 1008 amino acids and contains a cytoplasmic tail, a transmembrane domain, two calf domains, a thigh domain and a head (a β-propeller domain), which is the main ligand-binding domain (Figure 1) [2,15]. The β-subunit of αIIbβ3 consists of 772 amino acids and contains a cytoplasmic tail, a transmembrane domain, a membrane-proximal β-tail domain (βTD domain), four epidermal growth factor (EGF) domains, a hybrid domain and a β3A domain [15]. The β3 subunit consists of one polypeptide chain stabilised by disulphide bonds, which upon activation will rearrange through the formation of free thiols with a disulphide exchange reaction [16,17]. The integrin has multiple divalent cation-binding sites, necessary for binding of ligands. The cytoplasmic domains of both subunits form a ligand-binding domain for intracellular cytoskeletal molecules. Interaction of the α-subunit with the β-subunit is governed by the β-propeller domain and the β_3_A domain.

Nitric oxide and prostacyclin (PGI_2_), produced by the intact endothelium, inhibit the activation of discoid-shaped resting platelets [18,19]. During haemostasis, integrin αIIbβ3 plays a crucial role in platelet adhesion and platelet aggregation. Activation of platelets occurs when blood vessels are damaged. Platelets interact with collagen-bound von Willebrand factor (VWF) via the GPIb-V-IX receptor. Next to promoting platelet adhesion to the damaged vessel wall, VWF also enhances platelet aggregation under high shear conditions [20,21]. Following platelet activation, the content of alpha- and dense granules is secreted and by released adenosine diphosphate (ADP), platelet activation is further enhanced [22]. Upon platelet activation, conformational changes of the integrin take place, exposing the ligand-binding site of the fibrinogen receptor, which is unavailable in resting platelets, and as a consequence, fibrinogen crosslinks multiple platelets into aggregates, forming a platelet plug [23]

## 3. Ligands

The integrin αIIbβ3 has multiple ligands, categorised in ligands containing an RGD-motif (fibrinogen, fibrin, VWF, fibronectin) and other ligand-binding motifs, such as the KQAGDV-sequence from the fibrinogen γ-chain C-terminus, which is the primary binding site of fibrinogen [24,25].

## 4. Inside-out Signalling

Upon platelet activation and subsequent intracellular signalling, so-called “inside-out” signalling, integrin αIIbβ3 changes conformation (Figure 2). The integrin’s inactive state, in which the extracellular domain is folded with the ligand-binding site facing the membrane, transforms into an active state when the extracellular domain is extended (opened), thereby exposing the RGD ligand-binding domain with a high affinity for its ligands (Figure 1). In this high-affinity state, the cytoplasmic tails of the αIIb- and β3-subunits unclasp, separating the transmembrane domains [26].

Platelet activation induced by the binding of soluble agonists (e.g., ADP, thrombin, thromboxane A2) or adhesive ligands (e.g., collagen, VWF, fibrinogen) to their respective receptors, triggers signalling pathways involving key signalling proteins like phospholipase C (PLC), protein kinase C (PKC) and phosphatidylinositide-3-kinase (PI3K) [27]. These signalling events culminate in the activation of the small GTPase RAP1 (Ras-related protein 1), an important modulator of integrin αIIbβ3 activation. RAP1 is regulated by its activator CalDEG-GEFI, which responds to cytoplasmic Ca^2+^ rises, and RASA3, acting as a negative regulator of RAP1 activation. For sustained integrin activation, RASA3 activity needs to be restricted and this is accomplished by PI3K activation [28]. The effectors of RAP in platelets remain to be determined.

In addition to RAP1 activation, inside-out signalling leads to the recruitment and binding of actin cytoskeletal proteins talin and kindlin to the cytoplasmic tails of αIIbβ3. Through the separation of the talin head and tail domain, the talin head domain can associate with the β3-subunit cytoplasmic tail, which converts the integrin αIIbβ3 from the inactive to the active form [25]. The importance of kindlins in these integrin regulation processes is well established. Kindlin-1 is found in epithelial cells, kindlin-2 in solid mesenchymal tissue and kindlin-3 is found in hematopoietic cells. Cell studies revealed that the kindlins can only activate integrins together with talin [29].

## 5. Outside-in Signalling

Ligand binding to integrin αIIbβ3 triggers multiple complex signal transduction events in the cell, known as outside-in signalling, which promote actin polymerization and cytoskeletal reorganisation. Clustering of integrins by multivalent ligands further promotes outside-in signalling. Outside-in signalling drives important haemostatic processes as platelet spreading, stable thrombus formation and clot retraction (Figure 3). One of the earliest events occurring is tyrosine phosphorylation of different proteins by the Src family of kinases (SFK), c-Src, Lyn and Fyn. Following platelet activation, c-Src associates with the β3-integrin tail and becomes activated [25]. The work of Gong et al. showed that binding of the G protein subunit Gα_13_ to the β3-integrin tail promotes the activation of c-Src [30]. SFK activation then results in the phosphorylation and activation of the tyrosine kinase Syk and its interaction with the cytoplasmic tail of β3 [31]. Furthermore, SFK-induced phosphorylation of the β3 tail promotes the recruitment of intracellular adaptor proteins. Downstream of SFKs and Syk many signalling and cytoskeletal proteins become phosphorylated and/or activated, such as PKC, PLCγ2, focal adhesion kinase (FAK), PI3Kβ and RhoA [25]. PI3Kβ was shown to be of importance for thrombus stability [32,33], while RhoA regulates platelet spreading and clot retraction. Talin and kindlin-3 also appear to have a role in outside-in signalling by directly linking the β3 tail to the actin cytoskeleton [25,34].

## 6. Antithrombotic Agents from Nature

Abnormal platelet activation and aggregation can lead to thrombotic vessel occlusion [35]. Arterial thrombosis comprises the formation of platelet-rich thrombi under high shear flow as a result of eroded or ruptured atherosclerotic plaques, resulting in ischemic injuries. Arterial thromboembolism usually results in myocardial infarction or stroke, while venous thrombosis occurs under low shear flow around an intact endothelial wall and can lead to venous thromboembolism (VTE) and pulmonary embolism (PE) [36]. A difference between arterial and venous thrombosis is that venous thrombi constitute fibrin and red blood cells, with a few platelets, while arterial thrombi are platelet-rich [37]. Treatment of venous thrombosis usually consists of drugs directed against proteins involved in the coagulation cascade, while the treatment of arterial thrombosis is based on antiplatelet therapy [36]. However, several clinical studies have shown some beneficial effects on the combination of antiplatelet and anticoagulant therapies in patients with coronary and peripheral artery disease [38]. Some antiplatelet strategies are derived from nature.

Bloodsucking parasites such as mosquitoes, ticks, bugs, leeches, sandflies, hookworms and bats, express a vast variety of anticoagulant and antiplatelet proteins that counteract the host’s haemostatic system, allowing them to feed for extended periods of time. Therefore, these organisms represent major sources of lead compounds for the development of pharmacological tools and potentially useful therapeutic agents for the treatment of haemostatic disorders or cardiovascular disease. Feeding occurs through cannulation of venules or via haemorrhagic pools that accumulate in tissues because of a deep cut in the skin. One example of an anticoagulant peptide from a hematophagous parasite is hirudin. Hirudin (MW: 7 kDa) is isolated from the salivary glands of leeches and inhibits thrombin, by the formation of a complex of hirudin and thrombin, thereby inhibiting the activity of thrombin [39].

Snakes, especially vipers (venomous snakes), have venom in their glands that is rich in proteins that modulate blood clotting and thereby cause organ degeneration and generalised tissue damage. The function of these hemotoxic proteins is not only to immobilise the prey but also to aid in its digestion. Snake venoms have become a valuable source for drug development. One interesting group of peptides found in snake venom are the disintegrins.

Disintegrins are low molecular weight integrin-binding cysteine-rich peptides that have different functions, such as the inhibition of cell adhesion and proliferation. Most disintegrins have one of the following motifs: RGD or KGD, however, some have an MVD or RED motif. Most disintegrins display antagonistic activity towards integrins such as αIIbβ3, αvβ3, and α5β1, but also other integrins are inhibited. Disintegrins are found not only in snake venom but also in the excretion glands of hematophagous parasites. Three αIIbβ3-antagonising drugs are currently used in the clinic and two of them, eptifibatide and tirofiban, are derived from snake venom disintegrins. In contrast to eptifibatide and tirofiban which are low molecular weight compounds (see below), abciximab, the other FDA approved drug, is a chimeric monoclonal antibody binding to αIIbβ3.

In Table 1 an overview of αIIbβ3-targeting disintegrins from different species is given. Subsequently, we elaborate on specific examples per species, based on the number of hits on Pubmed.

## 7. Disintegrins from Snakes

### 7.1. Echistatin

Echistatin (5.4 kDa), first isolated from snake venom in 1988, is a polypeptide (49 amino acids) stabilised by 4 disulphide bonds [63]. Echistatin contains an RGD sequence and the half-maximal binding concentrations to several integrins were tested in solid-phase binding assays; integrin αvβ3 (0.46 nM), α5β1 (0.57 nM) and αIIbβ3 (0.90 nM) [63]. Echistatin is therefore not specific for the αIIbβ3 integrin. It is proposed that echistatin binds to the αIIbβ3 integrin to the same binding pocket on the β3 subunit as abciximab [100]. The clinically approved tirofiban has been developed from Echistatin and is discussed below.

### 7.2. Rhodostomin/Kistrin

Rhodostomin (8.4 kDa) consists of 68 amino acids and is stabilised by 6 disulphide bonds. Rhodostomin was expressed in *Pichia pastoris* and it inhibited cell adhesion of CHO cells (which express αIIbβ3 and αvβ3, IC_50_ 21 nM and 13 nM, respectively) to fibrinogen [61]. Rhodostomin also inhibited the adhesion of K562 cells to fibronectin (α5β1, IC_50_ 256 nM) [61].

## 8. Disintegrins from Ticks

### 8.1. Disagregin

Disagregin (6.9 kDa) consists of 60 amino acids stabilised by 3 disulphide bonds. Disagregin was isolated from the tick *Ornithodoros moubata* and inhibited ADP-stimulated platelet aggregation. Disagregin blocked platelet adhesion to fibrinogen, suggesting it is a fibrinogen antagonist [89]. Recently disagregin was obtained by total chemical synthesis and was shown to inhibit platelet adhesion, thrombus formation and fibrin formation [101].

### 8.2. Variabilin

Variabilin (5 kDa) consists of 47 amino acids and contains 5 cysteines. Variabilin was isolated from the tick *Dermacentor variabilis*, and inhibited ADP-induced platelet aggregation. Variabilin also blocked the interaction of αIIbβ3 with fibrinogen as well as the interaction of αvβ3 with vitronectin [93].

## 9. Disintegrins from Leeches

### 9.1. Decorsin

Decorsin (4.4 kDa) consists of 39 amino acids stabilised by 3 disulphide bonds. Decorsin was isolated from the leech *Macrobdella decora* and blocked the interaction of αIIbβ3 with fibrinogen and ADP-induced platelet aggregation [94]. Since leeches store blood for a long time, the antiplatelet aggregation function of this protein is important.

### 9.2. Ornatin E

Ornatin E (5.7 kDa) consists of 50 amino acids stabilised by 3 disulphide bonds. It is similar to decorsin. It was isolated from the leech *Placobdella ornate* and blocked the interaction of αIIbβ3 with fibrinogen (IC_50_ 4.4 nM) and platelet aggregation activated by ADP (IC_50_ 438 nM) [95]. Ornatin E was also recombinantly expressed, showing similar results as the isolated ornatin E, except showing a lower IC_50_ of ADP-induced platelet aggregation (284 nM) [102]. 

## 10. Disintegrins from Worms

### Hookworm Platelet Inhibitor (HPI)

Hookworm platelet inhibitor (15 kDa) was isolated from the worm *Ancylostoma caninum*. It inhibits platelet aggregation and platelet adhesion to fibrinogen (mediated via αIIbβ3) and collagen (mediated via α2β1) [97].

## 11. Disintegrins from Horseflies

### 11.1. Tabinhibitin

Tabinhibitin (25 kDa) was purified from the horsefly *Tabanus yao Macquart* salivary glands. It blocks platelet aggregation, suggested via the αIIbβ3 integrin, but it is not yet known whether tabinhibitin acts on other receptors [98].

### 11.2. Tablysin-15

Tablysin-15 (26 kDa) was purified from the horsefly *Tabanus yao Macquart* salivary glands. The sequence of Tablysin-15 is homologous to tabinhibitin. It blocks platelet aggregation and inhibits platelet adhesion to fibrinogen (αIIbβ3), endothelial cell adhesion to vitronectin (αvβ3) and endothelial cell adhesion to fibronectin (α5β1) [99].

## 12. Antagonist Binding to the αIIbβ3 Receptor; Used in (Pre)Clinical Setting

### 12.1. β3. Chain Binding

#### Abciximab

Abciximab (47.6 kDa) is the Fab fragment from a mouse/human chimeric antibody. When abciximab binds the β3 chain of the integrin receptor, the access of ligands to the binding pocket is impaired (Figure 4). Abciximab is not specific for the αIIbβ3 integrin, it can also block the vitronectin receptor αvβ3 with the same affinity (Kd 5 nM) and the MAC-1 receptor on leukocytes, albeit at lower affinity (Kd 160 nM) [103]. After administration, platelet function will restore within 24 to 36 h, however, Abciximab remains bound to platelets for 2 weeks after drug administration [104]. Although the main binding site of abciximab to αIIbβ3 is the ligand-binding pocket of the β3 subunit, it can also bind to KQAGDV, which is part of the αIIb subunit (Figure 4). Abciximab prevents access of ligands to binding pockets by steric hindrance or conformational effects [103]. Abciximab binds to both non-stimulated and stimulated platelets [105]. Abciximab is indicated for use in patients undergoing primary percutaneous coronary intervention and myocardial ischemia, together with heparin and acetylsalicylic acid [106]. Side effects of abciximab use are thrombocytopenia, bleeding, abdominal pain, nausea and vomiting [107]. Currently, abciximab is in shortage, as listed by the FDA, because of an interruption in manufacturing at a third-party site [108]. A schematic structure of abciximab is shown in Figure 5.

### 12.2. Binding Pocket between αIIb and β3 Subunits

#### 12.2.1. Eptifibatide

Eptifibatide (Integrilin; 832 Da) is a cyclic heptapeptide, containing a homo-arginine–glycine–aspartic acid (hArg–Gly–Asp) sequence, derived from the disintegrin barbourin (P22827), found in the venom of the southeastern pygmy rattlesnake (Sistrurus miliarius barbouri) [109,110]. The plasma half-life of eptifibatide is 3 h [111]. Cyclisation is usually performed to constrain the confirmation of a peptide, and it reduces the susceptibility to be degraded by proteases [112,113]. Eptifibatide is not specific for αIIbβ3 as it also binds to αvβ3, although with a lower affinity [114]. Eptifibatide is based on the KGD motif of barbourin, however, to enhance affinity the lysine was replaced by a homoarginine residue [50,110]. Eptifibatide can bind to both non-stimulated and stimulated platelets [110].

The binding site of eptifibatide is the binding pocket between the αIIb and β3 subunits, thereby blocking the binding domain for fibrinogen and thus inhibiting the formation of platelet thrombi (Figure 4). Because the binding site is in the ligand-binding pocket of αIIbβ3, eptifibatide competes with fibrinogen and VWF for the binding site, but not the binding site of abciximab [115]. Eptifibatide is used in patients with angina pectoris to prevent myocardial infarction and in patients undergoing primary percutaneous coronary intervention, often in combination with unfractionated heparin and acetylsalicylic acid [116]. Side effects of eptifibatide are bleeding, thrombocytopenia and hypotension [117]. The chemical structure of eptifibatide is shown in Figure 5.

#### 12.2.2. Tirofiban

Tirofiban (Aggrastat; 440 Da) is a synthetic non-peptide reversible antagonist of the αIIbβ3 platelet integrin. Tirofiban was designed to mimic the RGD recognition motif within disintegrin echistatin, isolated from the venom of the saw-scaled viper *Echis carinatrus* [62,110]. It can be cleared from the plasma by renal excretion, and its half-life is 2 h. Tirofiban has a higher affinity to the αIIbβ_3_ receptor compared to eptifibatide (Kd 15 nM) [103]. Tirofiban recognises the RGD-binding sequence and binds within the ligand-binding pocket of αIIbβ3 receptors to compete with fibrinogen and VWF (Figure 4). Tirofiban can bind to both non-stimulated and stimulated platelets [110]. Tirofiban is indicated in patients with acute coronary syndrome and in patients undergoing primary percutaneous coronary intervention, where its use is often combined with heparin [118]. Side effects of tirofiban are headaches, nausea, thrombocytopenia and bleeding [119]. The chemical structure of tirofiban is shown in Figure 5.

## 13. Oral Antagonists

In the past, various oral antagonists have been evaluated, such as orbofiban, sibrafiban, xemilofiban, lotrafiban and roxifiban, the use of which has been associated with a prolonged bleeding time, an increase in the incidence of thrombocytopenia and a 30–35% increase in mortality (including cardiovascular mortality) [120]. All the following oral antagonists are discontinued in clinical trials for arterial thrombosis, myocardial infarction and unstable angina pectoris. The chemical structures of these antagonists are shown in Figure 6.

### 13.1. Orbofiban

Orbofiban is a prodrug from SC-57101, a non-peptide antagonist, that blocks the binding of fibrinogen to αIIbβ3 thereby preventing platelet aggregation. After hydrolysis of the ethyl ester by esterases, the active SC-57101 prevents fibrinogen to bind to the receptor with an IC_50_ of 47 nM. SC-57101 binds to both resting and activated platelets (Kd; 70 nM and 109 nM, respectively). It can also reverse thrombus formation by binding to platelets even when fibrinogen is already bound [121]. In the OPUS-TIMI 16 trial, it was investigated whether orbofiban was effective. The trial arms were 50 mg of orbofiban twice daily, 50 mg of orbofiban twice daily for 30 days and thereafter 30 mg of orbofiban twice daily, or placebo (all groups received 150–162 mg of acetylsalicylic acid). However, after 30 days, the 50/30 treatment arm showed increased mortality [122]. Several groups performed research to investigate the mortality in the treatment arm, showing increased platelet reactivity [123] and neutrophil activation [124]. 

### 13.2. Sibrafiban

Sibrafiban is a prodrug from Ro 44-3888, a non-peptide selective antagonist of αIIbβ3 [125]. Platelet aggregation is blocked with IC_50_ 15 µg/L (~35 nM) [125]. In the SYMPHONY trial, it was investigated whether sibrafiban prevents more ischemic events than acetylsalicylic acid after acute coronary syndrome. The trial included three arms; acetylsalicylic acid (80 mg every 12 h), low-dose sibrafiban (3.0–4.5 mg every 12 h) and high-dose sibrafiban (4.5–6.0 mg every 12 h). Bodyweight and serum creatinine concentration were used to assign patients to the low- or high-dose group [126]. The result of the trial was that sibrafiban was as effective as acetylsalicylic acid for the prevention of ischemic events, however, it showed more bleeding in trial participants [127]. 

### 13.3. Xemilofiban

Xemilofiban is a prodrug from SC-54701, a non-peptide mimetic of RGDF, blocking the binding of fibrinogen to αIIbβ3. The active SC-54701 prevents fibrinogen to bind to the receptor with IC_50_ of 10 nM, showing increased potency when compared to SC-57101 [128]. In the EXCITE trial, it was investigated if long-term administration of xemilofiban prevents ischemic events and stabilises plaques. The trial included three arms; placebo, xemilofiban (10 mg, three times daily) and xemilofiban (20 mg, three times daily) [129]. The result of the trial was that xemilofiban (administration before PCI up until 6 months thereafter) did not decrease the incidence of death and nonfatal myocardial infarction. As a consequence of the lacking beneficial effect of xemilofiban, the development was discontinued.

### 13.4. Lotrafiban

Lotrafiban is a non-peptide prodrug from SB-214857 that blocks fibrinogen binding to αIIbβ3 [130]. Platelet aggregation is inhibited by lotrafiban with IC_50_ 40–80 nM [130]. In the BRAVO trial, it was investigated if lotrafiban therapy is effective in patients with recent unstable angina, stroke or myocardial infarction. The trial included two arms; placebo including acetylsalicylic acid, one time per day) or lotrafiban (30 mg or 50 mg, two times daily including acetylsalicylic acid, one time per day). The dose of lotrafiban was dependent on age and creatinine clearance. The phase III clinical trial was stopped because of increased mortality and increased thrombocytopenia and bleeding [131].

### 13.5. Roxifiban

Roxifiban (DMP754) is a prodrug of XV459, binds specifically to the αIIbβ3 receptor with IC_50_ 0.03–0.05 µM [132]. It binds to both non-stimulated and stimulated platelets [133]. In the ROCKET trial, it was investigated if roxifiban administration in patients with coronary artery disease affected platelet aggregation and receptor expression. The trial included three arms; acetylsalicylic acid (325 mg + placebo, one time per day), roxifiban (1.0 mg + placebo, one time per day) or the combination of acetylsalicylic acid and roxifiban (325 mg and 1.5 mg, one time per day). In clinical trials, roxifiban showed enhanced expression of P-selectin and PECAM-1. Changes in effectivity suggest that young platelets adjust their pathways to aggregate as roxifiban is still present and thus blocking the αIIbβ3 receptor [134].

## 14. Conclusions

For the prevention of thrombotic complications in patients suffering from acute coronary syndrome or unstable angina pectoris, the integrin αIIbβ3 antagonists abciximab, eptifibatide and tirofiban (all administered intravenously) are occasionally used in the clinical setting. Eptifibatide and tirofiban, based on the ligand (RGD)-mimetics, bind to the integrin with low affinity and induce a conformational change, which results in outside-in signalling and paradoxical platelet activation [135]. The use of all three inhibitors is complicated by the increased risk of bleeding and less frequently, thrombocytopenia. Thrombocytopenia is the result of an immunological reaction caused by antibodies directed against ligand-induced binding sites (LIBS), that become exposed after integrin/inhibitor interaction [135]. Typically, thrombocytopenia is more frequent with the use of abciximab, since more epitopes are exposed which can be recognised by antibodies already present in the bloodstream [136].

These ligand-mimetic effects might not be induced by disintegrins, since hematophagous parasites and snakes often have shielding surface components at the moment they become opsonized by the immune system of the host [137]. They can also change their surface to escape recognition by the host [137]. We can speculate that disintegrins have evolved into forms that escape the human immune system. A previous study revealed a disintegrin that did not cause a conformational change of integrin αIIbβ3, resulting in antiplatelet activity without causing bleeding [138]. More research into disintegrins, including structure-activity relationships, may provide more insight into their biological activity and mechanism of platelet inhibition.

Different oral integrin αIIbβ3 antagonists have been tested, but clinical trials were stopped due to undesired side effects, including excess mortality or the absence of improvements over existing therapy. Various arguments are described in the literature as to why these antagonists failed in clinical trials: (1) plasma concentrations vary over the treatment period, resulting in variation in levels of integrin inhibition and eventual thrombotic events [128]; (2) the bioavailability of oral antagonists results in differences in platelet inhibition in and between patients [139]; (3) the difference between early effects and chronic effects, suggesting that for chronic use, different therapeutic approaches are required. However, oral antagonists have some beneficial effects: the administration of drugs is easier, and patients can administer drugs themselves during chronic therapy. In addition, oral antagonists can be used for prophylaxis [140].

Antiplatelet therapy is the standard treatment of atherothrombotic disease in the heart, peripheral arteries and brain. Either monotherapy or dual antiplatelet therapy with acetylsalicylic acid (aspirin) and/or a P2Y12 receptor blocker (clopidogrel, prasugrel, ticagrelor or cangrelor) is recommended for the secondary prevention of arterial thrombotic events [141]. Acetylsalicylic acid is also used as prophylaxis, however, there is a controversy between international guidelines and research that has shown less benefit of prophylactic acetylsalicylic acid [142]. At this time point, αIIbβ3 inhibitors are only used in patients who undergo percutaneous coronary intervention (without pretreatment with P2Y12 receptor blockers) or in high-risk patients, because less bleeding complications occur with P2Y12 receptor blockers [143].

No new αIIbβ3 inhibitors are developed since the approval of current platelet antagonists. Research has shown already promising effects of αIIbβ3 inhibitors for the prevention of ischemia/reperfusion injury to preserve cardiac function [144]. Future directions could be based on structure-function analysis of exogenous disintegrins, which might aid in finding platelet antagonists that maintain haemostasis and have fewer adverse side effects.

## Figures and Tables

**Figure 1 ijms-22-03366-f001:**
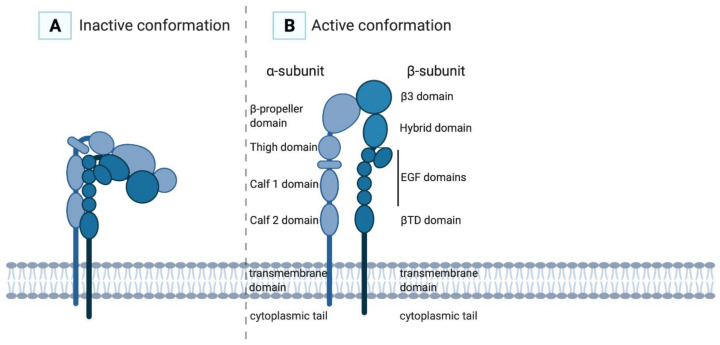
In the inactive conformation of the αIIbβ3 integrin (**A**), the ligand-binding site is poorly accessible for its ligands, while in the active conformation (**B**) the ligand-binding site is exhibited and has high affinity for its ligands.

**Figure 2 ijms-22-03366-f002:**
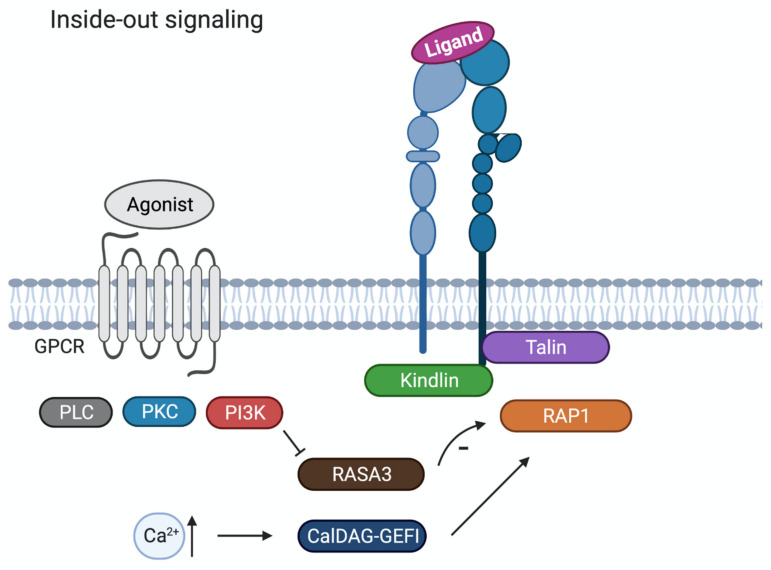
Schematic overview of inside-out signalling in platelets. Agonist activation of G protein-coupled receptor triggers signalling pathways with key signalling proteins like phospholipase C (PLC), protein kinase C (PKC) and phosphatidylinositide-3-kinase (PI3K). Increased Ca^2+^ will lead to activation of CalDAG-GEFI that will activate Ras-related protein 1 (RAP1). RASA3 acts as a negative regulator of RAP1 activation, and RASA3 activity is restricted by PI3K activation. Through the separation of the talin head and tail domain, the talin head domain associates with the cytoplasmic tail of the β3 subunit, converting the integrin αIIbβ_3_ from the inactive to the active form.

**Figure 3 ijms-22-03366-f003:**
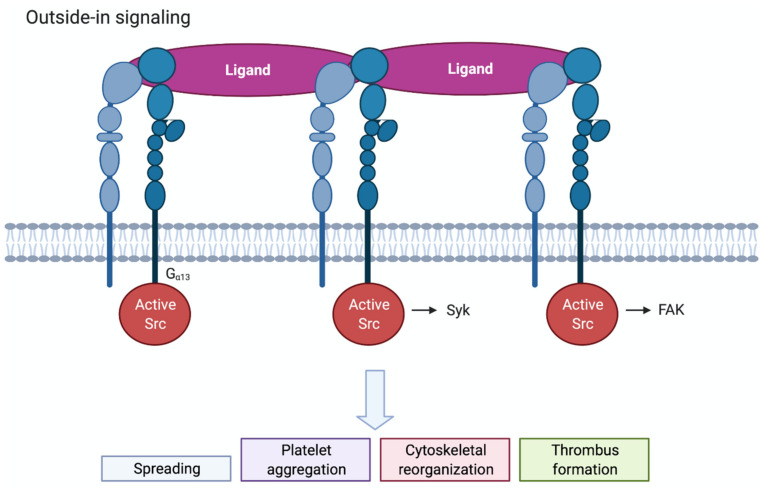
Schematic overview of outside-in signalling in platelets. Clustered integrins will initiate outside-in signalling. c-Src associates with the β3-integrin tail and becomes activated. Src, Syk and FAK will regulate downstream signalling via tyrosine phosphorylation. Outside-in signalling leads to spreading, platelet aggregation, cytoskeletal reorganisation and thrombus formation.

**Figure 4 ijms-22-03366-f004:**
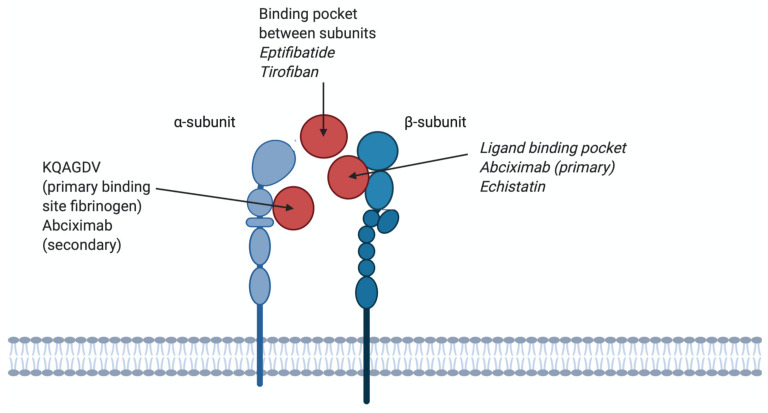
Different binding sites of antagonists on αIIbβ3 integrins. KQAGDV; the primary binding site for fibrinogen and a secondary binding site for abciximab. The binding pocket between subunits; binding site for eptifibatide and tirofiban, therefore competing with fibrinogen. Binding pocket β3; primary binding site for abciximab and echistatin, therefore not competing with fibrinogen.

**Figure 5 ijms-22-03366-f005:**
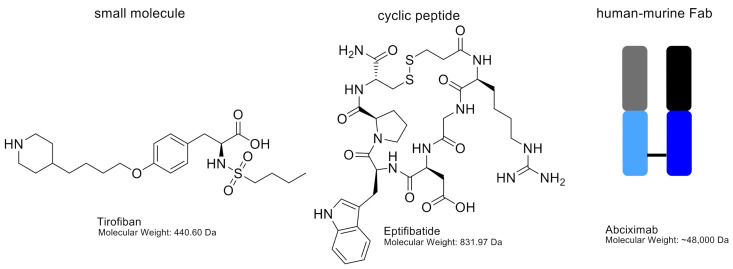
Chemical structures of tirofiban and eptifibatide, including molecular weights. Abciximab is shown as a schematic figure. The murine variable chain region is shown in grey (light chain) and black (heavy chain). The human constant regions are shown in light blue (light chain) and dark blue (heavy chain).

**Figure 6 ijms-22-03366-f006:**
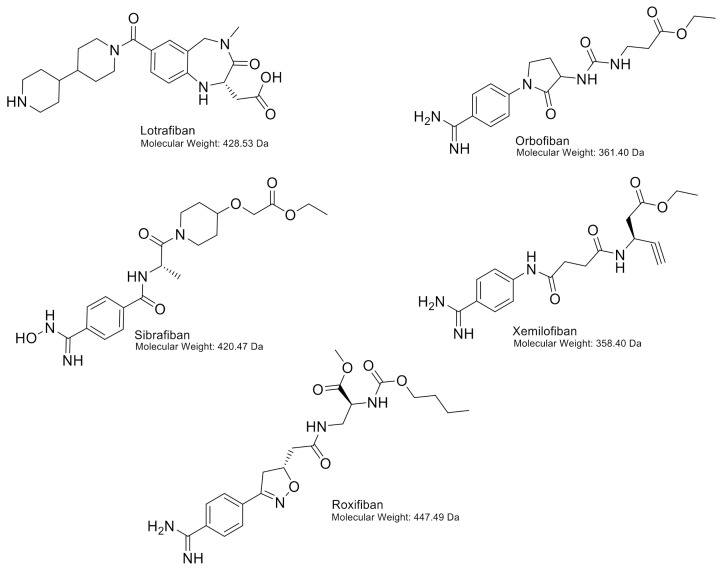
Chemical structures of lotrafiban, orbofiban, sibrafiban, xemilofiban and roxifiban, including molecular weights.

**Table 1 ijms-22-03366-t001:** Overview of the disintegrins in different species that target the αIIbβ3 integrin.

Name	Species	Target	Integrin	Sequence	Ref.
Accutin	Snake	platelets	αIIbβ3/αvβ3	RGD	[40,41]
Acostatin 1	Snake	platelets	αIIbβ3	RGD	[42]
Acostatin 2	Snake	platelets	αIIbβ3	RGD	[42]
Albolabrin	Snake	platelets	αIIbβ3	RGD	[43]
Albolatin	Snake	platelets	αIIbβ3/α2β1	KGD	[44]
Applagin	Snake	platelets	αIIbβ3	RGD	[45]
Arietin	Snake	platelets	αIIbβ3	RGD	[46]
Atrolysin E	Snake	platelets	αIIbβ3?/others?	MVD	[47]
Atropoimin	Snake	platelets	αvβ3	RGD	[48]
Atroxatin	Snake	platelets	αIIbβ3	RGD	[49]
Barbourin	Snake	platelets	αIIbβ3	KGD	[50]
Basicilin	Snake	platelets	αIIbβ3/αvβ3/α5β1	RGD	[51]
Batroxostatin	Snake	platelets	αIIbβ3	RGD	[52]
Bitistatin	Snake	platelets	αIIbβ3/αvβ3	RGD	[53]
Bothrasperin	Snake	platelets	αIIbβ3?/αvβ3	RGD	[54]
Cerastin	Snake	platelets	αIIbβ3/αvβ3	RGD	[51]
Cereberin	Snake	platelets	αIIbβ3/αvβ3	RGD	[51]
Colombistatin	Snake	platelets/human urinary and skin melanoma cancer cells	αIIbβ3/α5β1	RGD	[55]
Contortrostatin	Snake	platelets/cancer cells	αIIbβ3/αvβ3/ α5β1/αvβ5	RGD	[56,57]
Cotiarin	Snake	platelets	αIIbβ3/α5β1	RGD	[51]
Crotavirin	Snake	platelets	αIIbβ3	RGD?	[58]
Crotratroxin	Snake	platelets	αIIbβ3/α5β1	RGD	[51]
Cumanastatin 1	Snake	platelets	αIIbβ3	RGD?	[59]
Dendroaspin	Snake	platelets/endothelial cells	αIIbβ3/αvβ3/α5β1	RGD	[60,61]
Echistatin	Snake	platelets/osteoclasts	αIIbβ3/αvβ3/α5β1	RGD	[62,63]
Elegantin	Snake	platelets	αIIbβ3	RGD	[64]
Flavoridin	Snake	platelets	αIIbβ3	RGD	[65]
Flavostatin	Snake	platelets	αIIbβ3	RGD	[66]
Gabonin	Snake	platelets	αIIbβ3	RGD	[67]
Halysin	Snake	platelets	αIIbβ3	RGD	[68]
Halystatin	Snake	platelets	αIIbβ3?	RGD	[69]
Insularin	Snake	platelets/endothelial cells	αIIbβ3/αvβ3	RGD	[70]
Jararacin	Snake	platelets	αIIbβ3/αvβ3	RGD	[51,71]
Jarastatin	Snake	platelets	αIIbβ3	RGD	[71]
Jerdonatin	Snake	platelets	αIIbβ3	RGD	[72]
Lachesin	Snake	platelets	αIIbβ3/αvβ3/α5β1	RGD	[51]
Leucogastin A	Snake	platelets	αIIbβ3?	RGD	[73]
Leucogastin B	Snake	platelets	αIIbβ3?	RGD	[73]
Lutosin	Snake	platelets	αIIbβ3/αvβ3/α5β1	RGD	[51]
Mojastin 1	Snake	platelets/human urinary bladder cell adhesion to fibronectin	αIIbβ3/α5β1	RGD	[74]
Mojastin 2	Snake	platelets/human urinary bladder cell adhesion to fibronectin	αIIbβ3/α5β1	RGD	[74]
Molossin	Snake	platelets	αIIbβ3/αvβ3	RGD	[51]
Morulustatin	Snake	platelets	αIIbβ3?	?	[75]
Multisquamatin	Snake	platelets	αIIbβ3	RGD	[56,73]
Ocelatin	Snake	platelets	αIIbβ3?	RGD	[73]
PAIEM	Snake	platelets	αIIbβ3	RGD	[76]
Piscivostatin	Snake	platelets	αIIbβ3	RGD/KGD	[77]
Pyramidin A	Snake	platelets	αIIbβ3?	RGD	[73]
Pyramidin B	Snake	platelets	αIIbβ3?	RGD	[73]
Rhodostomin/Kistrin	Snake	platelets/endothelial cells	αIIbβ3/αvβ3/α5β1	RGD	[61]
Rubistatin	Snake	platelets/SK-Mel-28 cells	αIIbβ3?	MVD	[78]
Salmosin 1	Snake	platelets/endothelial cells	αIIbβ3/αvβ3	RGD	[79]
Salmosin 2	Snake	platelets	αIIbβ3	KGD	[80]
Saxatilin	Snake	platelets	αIIbβ3/αvβ3	RGD	[81]
Schistatin	Snake	platelets	αIIbβ3/αvβ3	RGD	[82]
Tergeminin	Snake	platelets	αIIbβ3	RGD	[50]
Triflavin	Snake	platelets	αIIbβ3	RGD	[83]
Trigramin	Snake	platelets/human melanoma cells	αIIbβ3	RGD	[84,85]
Trimucrin	Snake	platelets/endothelial cells	αIIbβ3/αvβ3	RGD	[86]
Ussuristatin-1	Snake	platelets	αIIbβ3	RGD	[87]
Ussuristatin-2	Snake	platelets	αIIbβ3/α5β1	KGD	[87]
Vicrostatin	Snake	platelets	αIIbβ3/α5β1	RGD	[88]
Viridin	Snake	platelets	αIIbβ3/αvβ3	RGD	[51]

Disagregin	Tick	platelets	αIIbβ3	RED	[89]
Ixodegrin	Tick	platelets?	αIIbβ3?	RGD	[90]
Monogrin	Tick	platelets	αIIbβ3	RGD	[91]
Savignygrin	Tick	platelets	αIIbβ3	RGD/RED	[92]
Variabilin	Tick	platelets	αIIbβ3/αvβ3	RGD	[93]

Decorsin	Leech	platelets	αIIbβ3	RGD	[94]
Ornatin E	Leech	platelets	αIIbβ3	RGD	[95]

Hookworm platelet inhibitor (HPI)	Worm	platelets	αIIbβ3/α2β1	KGD	[96,97]

Tabinhibitin	Horsefly	platelets	αIIbβ3	RGD	[98]
Tablysin-15	Horsefly	platelets/endothelial cells	αIIbβ3/αvβ3/α5β1	RGD	[99]

## Data Availability

Not applicable.

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
