# Peer review of "Exogenous Integrin αIIbβ3 Inhibitors Revisited: Past, Present and Future Applications"

_ijms, 2021, doi:10.3390/ijms22073366_

Round 1

Reviewer 1 Report

The authors made a lot of effort but I have a certain problem with the manuscript. There is quite a lot of review papers dealing with aIIbIII integrin and its inhibitors. Therefore, the focus of the present manuscript should be on reviewing of new potential drugs of animal origin rather than on multiplying  of information regarding structure and signalling of the integrin or drugs used in the clinic. If the authors still want to present a review on the latter aspects than the data on the drugs provided  in the paper are not sufficient.

Despite the title of the paper approx. half of its body is dedicated to aIIbIII integrin, its structure, and signalling. As long as description of the structure is important to explain the mechanism of action of the inhibitors, the review of signalling events do not seem to be that crucial for the main aim of the manuscript. Especially that there is a huge number of review papers dealing with this integrin.

In paragraph 7 (Disintegrins from snakes) eptifibatide has not been mentioned. In turn in paragraph 12 eptifibatide and tirofibane are both described . I don’t understand the reason for separation of these two paragraphs. Some of the information in these paragraphs as well as these in paragraph 6 (lines 175-187), are redundant, like for instance a species from which the substances were isolated. I would suggest to combine these paragraphs and also to add some more information regarding both eptifibatide and tirofibane: for instance chemical formulae  with the critical parts highlighted, history of medical use, medical indications for use, side effects, any data regarding non-responders or low responders due to integrin polymorphisms etc. The same data (except chemical formula) should be provided for abciximab.

Finally, the descriptions of some of the novel substances seem to be somewhat brief. For instance, while the data for orbofiban are given in details, information regarding the next four drugs is extremely concise. In the case of these drugs I would expect the same details as for orbofiban: doses, number of patients with side-effects etc. More detailed explanation of the proposed molecular mechanisms of the side-effects. This data are included in the papers to which the authors refer, but they mention them only briefly. It only serves as an appetizer, while it should be a main dish.  I would also expect the figures with chemical formulae of these potential drugs.

There are several inaccuracies in the paragraph 14 (Conclusions). Lines 364-367 regarding the use of P2Y12 inhibitors. Administration of clopidogrel and ticagrelor is not restricted to the patients after PCI or planned for PCI. Recommendations for the use of aspirin (or it should rather read acetylsalicylic acid, as aspirin is a brand name) in primary prevention is a matter of controversies, see 2019 ACC/AHA Guideline on the Primary Prevention of Cardiovascular Disease.

Minor points:

Wording in some phrases should be re-read and amended, for instance line 229 “the antiplatelet aggregation function”. Either antiplatelet or antiaggregatory.

Reviewer 2 Report

van den Kerkhof et al. present a review manuscript on the evolution of exogenous GpIIb/IIIa inhibitors. This manuscript has good potential, however, cannot be published as presented.  

  1. The sentences in lines 31-35 – citation needed. Since these sentences were heavily overviewed in Zhang et al. Cell Adh Migr. 2012, please cite that work.
  2. Citing Zhang et al. for the sentence in lines 37-38 is inappropriate. Cite the original work, not the review.
  3. Section 2 header – revise forward slashes for better readability as it may create a confusion to the reader.
  4. The sentence in line 57 – inappropriate citation.
  5. The sentence in line 58 – inappropriate citation. Cite Balduini’s work. Overall, all the citations need to be revised. Citations are intended to refer the reader to the original work, and not to the review paper briefly mentioning it.
  6. Restructure sentence in lines 73-74.
  7. This perception is wrong – treatment of arterial thrombosis consists of both antiplatelet and anticoagulant.
  8. Citation is needed in line 163.
  9. Revise sentence in 191.
  10. Please revise the CONCLUSION section. Readability is poor, ideas are stopped abruptly with sentences followed by discussing completely different aspects.
  11. Also, many statements have no experimental support, or simply wrong. For example, line 347 – is an inappropriate statement, thrombocytopenia can be seen with many antiplatelets, including all 3 clinically approved GPIIbIIIa inhibitors, both PDE inhibitors, as well as the first generation of P2Y12 inhibitors. Another one, “currently, the market is saturated with platelet inhibitors” – based on what do authors make this statement?
  12. Or line 368 – “GpIIbIIa inhibitors are used if thrombotic complications occur” – what about use of GpIIbIIIa inhibitors for the prevention of thrombotic complication? Or the use of other antiplatelets? I’d strongly recommend a revision of the whole by a person with clinical expertise in the field.

Reviewer 3 Report

The authors wrote a review on the αIIbβ3 integrin going from very basic aspects to clinical drugs developed to inhibit its functions. The manuscript is clear and well written, the structure is good. The part 12 about the (pre)clinical use of αIIbβ3 inhibitors would gain comprehensibility with the presence of a table summarizing the different drug actions, development, and status in the clinical environment.

Minor: the recommended ISTH abbreviation for von Willebrand factor is VWF.

Reviewer 4 Report

Review of manuscript ijms-1124478 entitled: Exogenous integrin αIIbβ3 inhibitors revisited: past, present and future applications  by Danique L. van den Kerkhof , Paola E.J. van der Meijden , Tilman M. Hackeng , Ingrid Dijkgraaf

The manuscript is the interesting and comprehensive review devoted to inhibitors of integrin αIIbβ3 in antiplatelet and anticoagulant therapy. The paper contains clear introduction discussing both structure and function of integrins. Next chapters are focused on structure and function of integrin αIIbβ3 or explanation of inside-out and outside-in signaling. The disintegrins derived from different species of bloodsucking parasites were also discussed. The clinical problems of integrin αIIbβ3 antagonists administration (orally administered drugs including) were also described. The final chapter encapsulates the main problems discussed within the manuscript.

The manuscript could be interesting for the readers of International Journal of Molecular Sciences.

There are a few concerns with the study:

Major concerns

  1. The paper contains scarce clinical data that would be interesting for the reader. The main idea of the disintegrin selection for the clinical use is not very well described. What is the mechanism of mortality caused by some oral antagonists. Are there more unspecific (unwanted) effects, what is the pathogenesis of these effects?
  2. The future directions of the research devoted to discussed antiplatelet medicines should be highlighted.

Minor concerns

  1. On page 3 (lines 98-106) the figure 2 should be cited on the beginning of the paragraph. It will be easier to comprehend the text keeping track of figure.
  2. Some examples of antagonists derived from parasites were described in details. What was a reason of selection?

Round 2

Reviewer 1 Report

The structure of some part of the manuscript is still not clear to me. Paragraph 6  is entitled Imbalanced platelet aggregation. Why? Most of the paragraph deals with antithrombotic agents produced by animals and information regarding imbalanced platelet aggregation is limited to few lines. The content of this paragraph is also quite chaotic: eptifibatide and tirofiban are disintergrins or are derived from disintegrins, but the  idea of disintegrins is explained later (lines 194-201) than these drugs are mentioned for the first time (191-192).  Why abciximab appears in this context?

The author’s explanation of why a snake disintegrin eptifibatide is not described in a paragraph entitled “Disintegrins from snakes” is unconvincing. According to this explanation a correct title of the paragraph should read: “ 2 snake venom disintegrins per species, based on the amount of hits on Pubmed”. I simply don’t understand the reason for the paragraph 7. Echistatin is discussed in 12.2.2. and kistrin shoud be also moved to paragraph 12.

The rest of the manuscript has been improved.

Author Response

Authors reply: We thank the reviewer for these comments. We changed the title of paragraph 6 into: antithrombotic agents from nature.

We also changed the order in paragraph 6. Disintegrins are first introduced and thereafter eptifibatide and tirofiban are introduced. We introduced abciximab in this paragraph, because we mention that ‘Three αIIbβ3-antagonizing drugs are currently used in the clinic’. Abciximab is therefore named to complete the αIIbβ3-antagonizing drugs. Further in the review, we expand the information about abciximab.

As suggested by the reviewer, we added the sentence: ‘.. based on the number of hits on Pubmed’. (line 212-213) to inform the readers where we based our decision on. Eptifibatide itself is not a disintegrin, however barbourin, where eptifibatide is derived from, is. Echistatin and Rhodostomin/kistrin (both disintegrins from snakes) showed more hits on Pubmed then Barbourin did. We chose to give a complete overview of the currently used inhibitors, including eptifibatide. We agree with the reviewer that echistatin is shortly discussed in 12.2.2, however we chose to write paragraph 7 to elaborate on the most discussed disintegrins from different species, and echistatin is one of them.

Reviewer 2 Report

English editing is required. 

Author Response

Authors reply: We thank the reviewer for this comment. We are willing to send our review to an official company for English editing.

Reviewer 4 Report

Thank You very much for revision.

Author Response

Authors reply: We thank the reviewer for this comment.